# Ultrafast population coding and axo-somatic compartmentalization

Chenfei Zhang[1,2,3☯], David Hofmann [ID][1,2,3,4,5☯], Andreas Neef [ID][1,2,3,6,7]*,
Fred Wolf [ID][1,2,3,6,7,8]*

**1** Max Planck Institute for Dynamics and Self-Organization, Göttingen, Germany, **2** Göttingen Campus Institute for Dynamics of Biological Networks, Göttingen, Germany, **3** Bernstein Center for Computational Neuroscience, Göttingen, Germany, **4** Department of Physics, Emory University, Atlanta, Georgia, United States of America, **5** Initiative in Theory and Modeling of Living Systems, Emory University, Atlanta, Georgia, United States of America, **6** Institute for the Dynamics of Complex Systems, Georg-August-University Göttingen, Göttingen, Germany, **7** Max Planck Institute of Experimental Medicine, Göttingen, Germany, **8** Center for Biostructural Imaging of Neurodegeneration, Göttingen, Germany

☯ These authors contributed equally to this work.
* aneef@gwdg.de (AN); fred.wolf@ds.mpg.de (FW)

**Data Availability Statement:** The code used to produce the data is available at the github repository https://github.com/chenfeizhang/Brette_gwdg All relevant data are within the manuscript and its Supporting information files.

## Abstract

Populations of cortical neurons respond to common input within a millisecond. Morphological features and active ion channel properties were suggested to contribute to this astonishing processing speed. Here we report an exhaustive study of ultrafast population coding for varying axon initial segment (AIS) location, soma size, and axonal current properties. In particular, we studied their impact on two experimentally observed features 1) precise action potential timing, manifested in a wide-bandwidth dynamic gain, and 2) high-frequency boost under slowly fluctuating correlated input. While the density of axonal channels and their distance from the soma had a very small impact on bandwidth, it could be moderately improved by increasing soma size. When the voltage sensitivity of axonal currents was increased we observed ultrafast coding and high-frequency boost. We conclude that these computationally relevant features are strongly dependent on axonal ion channels' voltage sensitivity, but not their number or exact location. We point out that ion channel properties, unlike dendrite size, can undergo rapid physiological modification, suggesting that the temporal accuracy of neuronal population encoding could be dynamically regulated. Our results are in line with recent experimental findings in AIS pathologies and establish a framework to study structure-function relations in AIS molecular design.

## Author summary

In large nervous systems, a signal often diverges to hundreds or thousands of neurons. This population's spike rate can track changes in this common input for frequencies up to several hundred Hertz. This ultrafast population response is experimentally well established and critically impacts cortical information processing. Its underlying biophysical determinants, however, are not understood. Experiments suggest that the ion channels at the axon initial segment strongly contribute to the ultrafast response, but recent

**Funding:** This work was supported by the China Scholarship Council (to CZ), the United States National Institute of Health (NIH) BRAIN Initiative Theory Grant 1R01-EB022872 (DH), and NIH grant 5R01-NS099375 (DH), the Bundesministerium für Bildung und Forschung (BMBF, Federal Ministry of Education and Research) grant 01GQ1005B (FW, AN), the VolkswagenStiftung under grant no. ZN2632 (to FW), and through the Deutsche Forschungsgemeinschaft (DFG, German Research Foundation) through CRC 889 and 1286 (FW) and through grant 436260547 (AN, FW), in relation to NeuroNex (NSF 2015276). The funders had no role in study design, data collection and analysis, decision to publish, or preparation of the manuscript.

**Competing interests:** The authors have declared that no competing interests exist.

theoretical studies emphasize the importance of neuron morphology and the resulting resistive coupling between axon and somato-dendritic compartments. We provide an exhaustive analysis of the population response of a simplified multi-compartment model. We vary the axo-somatic interaction and also active axonal properties and compare models at equivalent working points, avoiding bias. This approach provides a guideline for future experimental and theoretical studies. In this framework, the population response is closely associated with the AP generation speed at the AP initiation site, which is mostly determined by axonal ion channel voltage sensitivity. The resistive axo-somatic coupling has an additional modulatory influence. These insights are expected to hold for encoding mechanisms of more sophisticated models, suggesting that physiological changes to axonal ion channels could modulate the population response rapidly.

## Introduction

Humans, monkeys, and other mammals can perform complicated cognitive tasks that engage deep cortical hierarchies with processing times as brief as a few hundred milliseconds [1, 2]. To achieve its overall speed, such fast information processing presumably depends on an ultrafast response dynamics of local cortical populations. Over the past years, numerous experimental studies tested different populations of cortical neurons for their capability to perform ultrafast population coding [3–15]. These studies used time domain and frequency domain analyses to assess the dynamics of the population's mean firing rate in response to a shared input superimposed on a continuously fluctuating background, mimicking ongoing synaptic input. Using a time domain approach, it was shown for instance that populations of layer 2/3 pyramidal neurons in visual cortex nearly instantaneously respond to a small step in the average input [7]. In the frequency domain, the capability to generate an ultrafast response can be characterized by the population's dynamic gain function. Dynamic gain was introduced theoretically by Bruce Knight in 1972 [16] and measures the dynamic population rate response to sinusoidal input signals embedded in a background of i.i.d. statistically stationary noise. The frequency-domain correspondence of a sub-millisecond, ultrafast response is a dynamic gain function that starts to decay only for input frequencies above several hundred Hz. In cortical pyramidal cells, dynamic gain was first measured for populations of layer 5 pyramidal neurons, which indeed display a near-constant response up to a few hundred Hz [4], see also [3] for an earlier indication of high bandwidth. As experimentally assessed, the dynamic gain function and the ultrafast response are not fixed and invariant properties of a neural population but substantially depend on the nature of background activity. Confirming a seminal theoretical prediction [17], when the correlation time of the background input is increased, the dynamic gain in the high-frequency regime is substantially boosted [7], a phenomenon previously dubbed the Brunel effect [7].

At first sight, the generation of ultrafast responses by an asynchronous population of neurons and its dependence on background fluctuations appear easy to understand. In the absence of external input, the cells fire asynchronously, because their background inputs are not correlated. This asynchrony guarantees that at any moment some fraction of the population is close to threshold and thus is in principle capable of an immediate, near instantaneous response. Intriguingly, however, several experimental studies demonstrated that a population's propensity to generate an ultrafast response sensitively depends on subtle aspects of the cells' biophysical and molecular design. In particular the molecular architecture of the spike initiation zone in the axon initial segment (AIS) seems to be specifically tailored to support an ultrafast

response. Lazarov et al. recently demonstrated that mutation of AIS-specific cytoskeletal proteins specifically impairs population response speed even if the ability of the cells to generate spikes per se is essentially unaffected [14]. Consistent with this observation, the bandwidth of pyramidal neurons is substantially reduced after recovery from pathological conditions, such as transient hypoxia that affect AIS molecular architecture [18]. Both of these findings are in line with prior studies suggesting that immature AIS organization [13] or pharmacological blockade of sodium channels at the AIS [9] disrupt dynamic gain bandwidth and ultrafast population response. While the cellular and biophysical features sufficient to equip a cortical population with an ultrafast response are not completely understood, the weight of experimental evidence strongly suggests that this population-level property is sensitive to slight alterations in single neuron biophysics. It is interesting to note that phylogenetic studies place the response-speed-related refinements of AIS architecture at the invention of forebrain population coding in the first large chordate brains [14, 19, 20]. A series of intriguing studies [10, 21–23] have even implicated inter-individual differences in the capability of human cortical neurons to support ultrafast population encoding as a determinant of heritable inter-individual difference in cognitive performance.

The feasibility of ultrafast population coding, its dependence on the structure of background input correlations, and its sensitivity to single neuron biophysical and molecular design have been correctly predicted by theoretical work preceding this recent wave of experimental studies. Theoretical studies, in single-compartment models, showed that ultrafast population encoding is closely associated with active AP initiation dynamics [9, 24–29]. Neuron models that model AP generation by threshold-crossing, tacitly assuming an infinitely fast AP initiation, were first shown to exhibit a dynamic gain function with high bandwidth and the Brunel effect [17]. In subsequent studies [24–26], the voltage dependent sodium currents underlying AP initiation were incorporated. High bandwidth encoding could only be recovered when the voltage sensitivity of sodium currents was very high [24]. Based on these observations, the properties of the ion channels, which control the rapidness of AP initiation, were hypothesized as key parameters determining the bandwidth of the dynamic gain function [27, 30].

The experimental studies summarized above suggest such a dependence in two ways. Firstly, the onset rapidness of the somatic AP waveform and the dynamic gain bandwidth measured in individual cells were found to be positively correlated [13, 14]. Secondly, when the sodium channels at the AIS were blocked by TTX [9] or when the number of ion channels at the AIS was reduced by genetic manipulation [14], the bandwidth of the dynamic gain function and the AP onset rapidness are both reduced. A distinct but related mechanism has been suggested to operate in multi-compartment models that describe the spatio-temporal dynamics of AP generation along the somato-axonal axis. APs are generated in the AIS, spatially separated from the soma, and lateral currents between these compartments are important for the detailed dynamics of AP initiation. Brette and coworkers [31, 32], formulated an idealized multi-compartment model to study the impact of AP initiation site on somatic AP waveform. Increasing the distance between AP initiation site and soma, they predicted a discontinuous increase of the lateral current to emerge during AP generation. Even with a low intrinsic voltage sensitivity of the sodium currents underlying AP initiation, they suggested the conditions for an ultrafast population response might be fulfilled. A third hypothesis focuses on the impact of single neuron dendritic trees on population encoding [11, 33, 34]. Eyal et al. [33] first showed that multi-compartment models of the entire dendrite-soma-axon axis can realize high bandwidth encoding with a cutoff frequency above 100Hz, in particular in the presence of a large dendritic compartment. Intriguing support for this specific mechanism was found in a modeling and experimental study of Purkinje cells that exhibit an exceptionally large

dendritic tree and show a very high bandwidth population response [11]. All of these biophysical mechanisms, only a subset, or additional mechanisms yet unknown may underlie the presence or absence of ultrafast response dynamics in different neuronal populations. Irrespective of which of these alternatives ultimately apply, the increasing specificity with which recent experimental studies are probing the biophysical basis of ultrafast population coding calls for a systematic and controlled methodology to examine neuronal design variants at equivalent operating points and analyze the resulting impact on dynamic gain functions.

Here, we introduce such a methodology and use it to provide a dynamic gain analysis of the idealized multi-compartment model established by Brette [31]. In particular, we examine the impact on population response dynamics of various passive and active biophysical parameters representing AIS structure. To precisely compare the dynamic gain functions of different model variants at equivalent operating points, the mean and standard deviation of the inputs were chosen to maintain fixed average firing rate and coefficient of variation (CV) of the inter-spike interval (ISI) distributions. Stimulus filtering between the soma and the AP initiation site and the contribution of lateral currents to AP initiation are two fundamental differences between Brette's model and simpler single compartment models. We find that electrotonic separation between AIS and soma and stimulus filtering does not, per se, result in the emergence of an ultrafast population response. Increasing the voltage sensitivity of sodium current induces high bandwidth and a sensitivity of dynamic gain to the input correlation (Brunel effect), demonstrating that this simple model is in principle capable of ultrafast encoding. In contrast, increasing the sodium conductance had no substantial impact on the dynamic gain function. Our results show that, in this idealized model, the somatic AP waveform has little relation to the emergence of an ultrafast population response. In contrast, the key determinants of ultrafast population coding are the voltage dependence of axonal sodium currents and the AP waveform at the AP initiation site around the time of AP onset.

## Results

### Voltage decoupling between soma and axon under voltage clamp and current clamp

We investigated the multi-compartment model introduced in [31] (Fig 1A) implemented in NEURON (see Methods). Our implementation reproduced the central feature reported by Brette [31]: when the somatic voltage is clamped and changed so slowly that all currents attain their steady-state values, then we can observe a bifurcation in the voltage difference between soma and axon (dashed line in Fig 1C). The mechanism behind this bifurcation is explained Fig 1B. If the somatic voltage is fixed, the model is in steady state, if the sodium current equals the lateral current. This corresponds to the intersection points between the sodium current and the lateral current as a function of axonal voltage. The total resistance between the soma and the position of the sodium channels increases with $x_{Na}$ and shapes the slope of the lateral current function. For $x_{Na} = 20\mu m$, increasing the clamped somatic voltage from −60mV to −50mV smoothly changes the intersection point of the two functions. However, when the position of sodium channels from the soma exceeds a critical distance, about $27\mu$m (Fig 1B middle panel), the number of intersection points increases to three. Thus, under the artificial condition of a fixed somatic voltage, a sufficiently large distance between initiation site and soma leads to a bifurcation of the axonal equilibrium voltage under a slow, quasi-stationary increase of the somatic voltage. This results in an abrupt change of the lateral current entering the soma once the somatic voltage crosses a threshold value. This effect is a candidate mechanism to generate sharp APs at the soma and was hypothesized to induce ultrafast population encoding in cortical neurons [31].

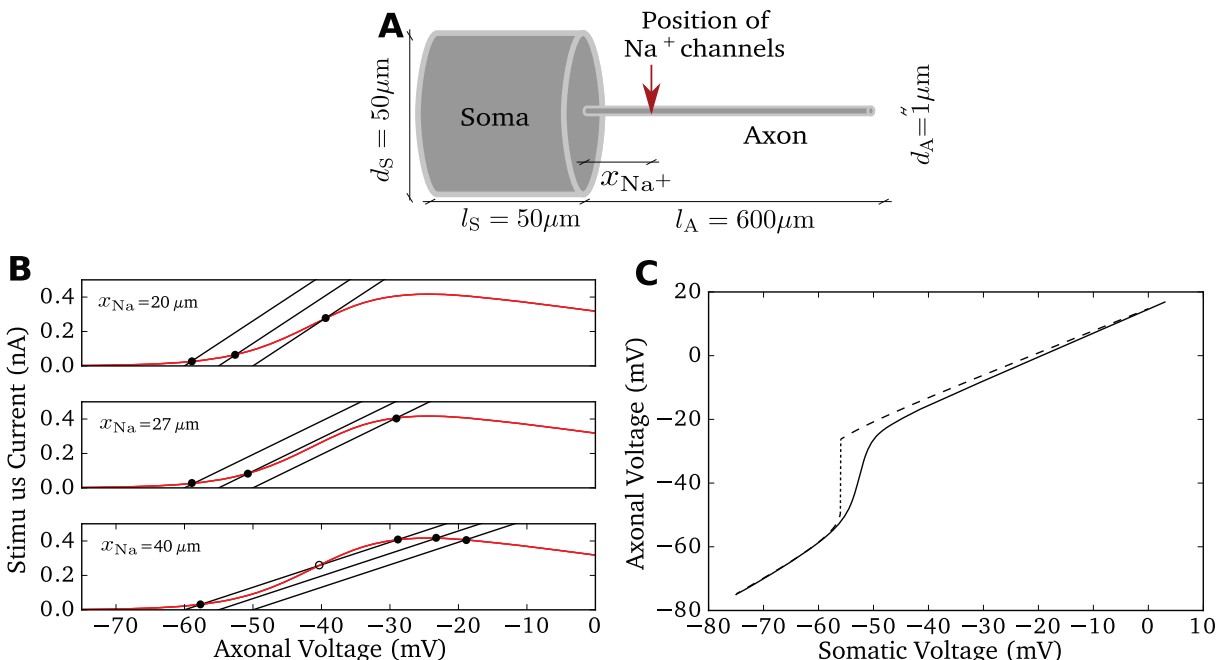

**Fig 1. Model properties. A** Sketch of the model morphology: soma and axon are modeled as cylinders. The sodium conductance is positioned at $x_{Na}$ and the stimulus current is injected at the middle of the soma. For simulation and model details see Methods. **B** Bifurcation of axonal voltage, replicating Fig 2 of [31]. Sodium current (red) and lateral currents (black, with somatic voltage fixed at −60mV, −55mV, −50mV respectively) are plotted as a function of axonal voltage at different $x_{Na}$. The intersection points between the black curve and the red curve indicate stationary axonal voltage given the somatic voltage. For large $x_{Na} = 40\mu m$, the first intersection point changes discontinuously for increasing somatic voltage. **C** Axonal voltage plotted against somatic voltage for two conditions with $x_{Na} = 40\mu m$. Only when the somatic voltage is artificially fixed to a slowly increasing potential (dashed), a sudden jump in the axonal voltage occurs (dotted), when the voltage control of the axonal voltage is lost around −55mV. In a dynamic situation, here a current clamp with injection of a constant current driving firing at 5 Hz, this jump disappears (solid line), the transition is more gradual.

In Fig 1C, we plot the axonal voltage as a function of somatic voltage for $x_{Na} = 40\mu m$. With the somatic voltage clamped to different values, we recorded the corresponding stationary axonal voltages. As expected, there is a discontinuity around the somatic voltage of −55mV, the somatic voltage at which two intersections between lateral and sodium current meet. This corresponds to the annihilation of a stable and an unstable fix point. However, it is important to note that the abrupt change of the voltage difference in Fig 1B must not be confused with a very rapid change of the axonal voltage or a high voltage sensitivity of the change of the axonal voltage. Only the artificial construct $d\left(\frac{dV_{axon}}{dt}\right)/dV_{soma}$ is very large. The axonal phaseplot's slope $d\left(\frac{dV_{axon}}{dt}\right)/dV_{axon}$ is not necessarily drastically changed at this bifurcation point.

Apart from simulations with a voltage clamped soma, we also studied the current clamp condition, in which we injected a slowly increasing input current in the middle of the soma, and recorded the two voltage traces at the soma and at the AP initiation site. In this case, the lateral current from axon to soma not only slows axonal depolarization, but accelerates the depolarization of the soma. As shown in Fig 1C (continuous line), the decoupling of somatic and axonal voltages was substantially smoother in this dynamic case. Even if the parameter $x_{Na}$ is sufficiently large for the bifurcation under the voltage clamp case, we only observed a gradual deviation of the two voltages. The deviation speed is limited by the activation speed of the sodium current. The maximum amplitude of the sodium current remains unaffected, so for larger voltages, the decoupling distances of the two voltages are maintained.

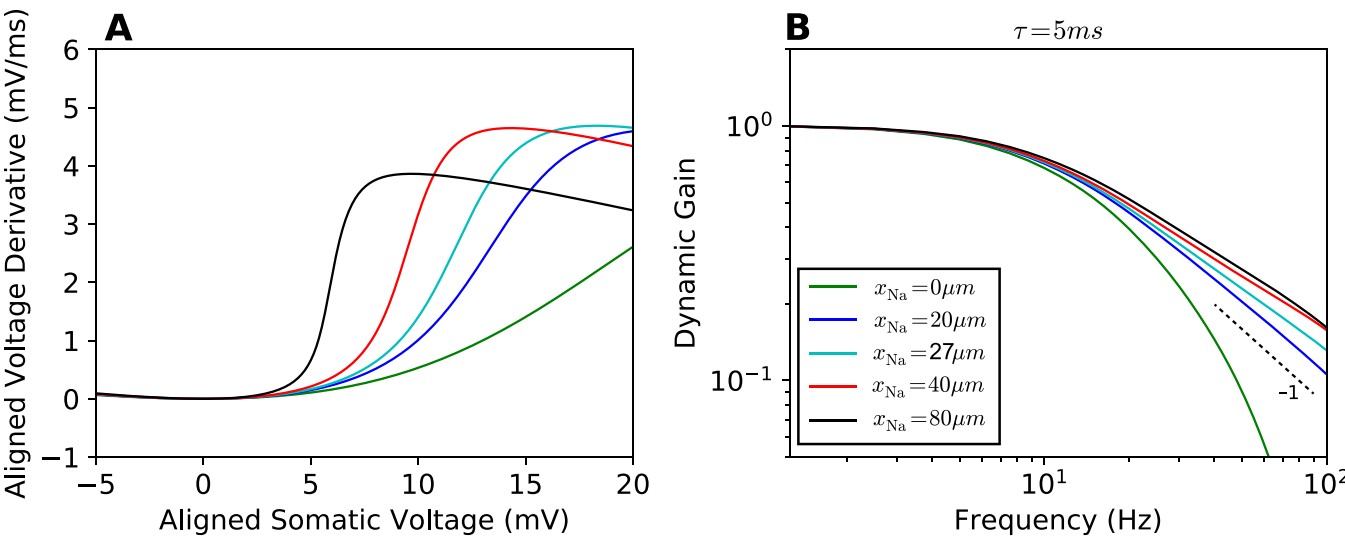

**Fig 2. Phase plot and linear response functions. A** Phase plots for the APs seen at the soma, somatic voltage rate of change vs. somatic voltage for different sodium channel positions $x_{Na}$ (legend in **B**). The local minima of all curves are aligned at (0mV, 0mV/ms). **B** Dynamic gain functions for different $x_{Na}$, normalized. 95% confidence intervals are plotted as shaded areas, mostly hidden by the line width. All curves are above statistical significance threshold (see Methods). The input current correlation time was set to $\tau = 5$ms.

## Spatial decoupling is not sufficient for ultrafast population encoding

Next, we characterized the sharpness of the somatic action potential as we varied the sodium channel position $x_{Na}$ from $0\mu$m to $80\mu$m (Fig 2A). We found that the somatic voltage rose more rapidly, the further away from the soma the sodium channels were positioned. The AP initiation dynamics at $x_{Na} = 27\mu$m did not exhibit a bifurcation.

We then calculated the dynamic gain for various values of the position parameter $x_{Na}$ as depicted in Fig 2B. It is important to note that the model parameters do not fully determine the dynamic gain function. Instead, the working point, at which the model is studied, influences the dynamic gain function. In order to provide an unbiased comparison of different models, we always selected the current's average and standard deviation to achieve a firing rate of 5 Hz and a CV of ISI of 0.85, unless noted otherwise. In every case, spikes from 400,000 seconds of simulation were used to calculate a dynamic gain function. The correlation time of the fluctuating background current was set to 5ms. Moving the sodium channels away from the soma somewhat enhanced the high-frequency response, in particular for $x_{Na} = 0$ to $x_{Na} = 20\mu$m. The distance for which a bifurcation in the voltage-clamp condition was observed does not stand out in any way. Moreover, all obtained dynamic gain curves exhibited a cutoff frequency of about 10Hz and the dynamic gain curves converged in a form similar to a canonical conductance-based neuron model with a decay exponent of -1 [24]. Increasing $x_{Na}$ from $40\mu$m to $80\mu$m had limited effect on the dynamic gain curve, despite doubling the electrotonic separation between AP initiation site and soma.

Compared to a single compartment model, there are two main differences in fluctuation-driven AP initiation for the current multi-compartment model. Firstly, in a single compartment model, the injected stimulus waveform directly enters the dynamic equation of AP generation. In the multi-compartment model, the spatial separation between injection site and initiation site creates a filter. The cable properties attenuate the stimulus as it is transmitted to the AP initiation site. This may decrease the dynamic gain in the high-frequency range relative to a single compartment model. Secondly, the current components underlying AP generation

are modified. In a single compartment model the entire local sodium current is effective for AP initiation. In the multi-compartment model, the sodium current, entering at the AP initiation site, not only charges the local membrane capacitance, but also feeds lateral currents that first depolarize neighboring parts of the axon and subsequently the somatic membrane. This lateral current from the AP initiation site depends not only on the distance to the soma, but also on the axial resistivity of the axon. Thus the axial resistivity is an additional parameter that impacts on the voltage dynamics at the AP initiation site of a multi-compartment model.

The lateral current away from the AP initiation site slows down the dynamics of AP generation. Moving the AP initiation site away from the soma (Fig 2A), we found that the shape of the APs at the soma is changed, but the AP initiation dynamics in the axon is modified much more mildly. Fig 3A and 3B show the impact of lateral current and transfer impedance on axonal voltage dynamics and population encoding. In Fig 3A, the phase plot for APs at the AP initiation site shows that shifting the AP initiation site away from the soma increases the resistance between, thus reduces the lateral current towards the soma during the AP upstroke. For larger $x_{Na}$ values, the AP initiation dynamics thus becomes more rapid. Fig 3B depicts the filtering effect due to stimulus transmission from the soma to the AP initiation site. To obtain these transfer impedance curves, we injected fluctuating currents obtained from Ornstein-Uhlenbeck processes with a correlation time of 5 ms. Average current and standard deviation were the same as we later used to calculate the dynamic gain. We used the axonal voltage at $x_{Na}$ as the continuous response variable and calculated the transfer function from somatic input current to axonal voltage at $x_{Na}$. The dominant effect is the filtering from current to somatic voltage. With increasing $x_{Na}$, a small additional damping of high-frequency components appeared. Below 100Hz, this additional damping effect was weak for $x_{Na} = 20\mu m$, $40\mu m$, $80\mu m$. These results suggest that the dynamic gain values are enhanced for larger $x_{Na}$ because the larger axonal resistance reduces the lateral current while the input experiences only minor damping. Further increasing $x_{Na}$, one expects that the AP initiation dynamics saturates while the transfer impedance will eventually reduce the high-frequency gain. This should lead to an impaired encoding ability for high frequencies. Fig 3C confirms this prediction. We calculated the dynamic gain function for $x_{Na} = 200\mu m$. The dynamic gain was reduced compared to that for $x_{Na} = 80\mu m$. Another expectation is that by reducing the lateral current, the dynamic gain functions should become more sensitive to transfer impedance with larger $x_{Na}$. To examine this, we increased the axial resistivity $R_a$ fivefold to $750\Omega cm$. The dynamic gain functions in Fig 3D show that with lateral current suppressed, increasing $x_{Na}$ reduces the high-frequency dynamic gain monotonically opposite to the behaviour shown in Fig 2B.

## Negligible impact of sodium peak conductance on dynamic gain

Experimental studies have shown that a high density of sodium channels in the AIS is required for high bandwidth encoding [9, 14]. We thus investigated how changing sodium peak conductance impacts on population encoding in the multi-compartment model. The sodium peak conductance was increased 5 and 10 fold. Fig 4A and 4C shows the dynamic gain curves for 1Hz and 5 Hz firing rate. The CV was controlled to vary less than 0.1 for both rates. Increasing the sodium peak conductance caused only minor changes to the dynamic gain curves. For the 1 Hz firing rate, the dynamic gain at high frequencies was slightly enhanced for higher sodium peak conductance. Further increasing the sodium peak conductance to $10\bar{g}_{Na}$, the gain curve remained almost identical to that at $5\bar{g}_{Na}$. For 5Hz firing rate, the encoding of high frequencies was overall better than for the 1 Hz condition. However, increasing the sodium peak conductance led to a slight decrease of dynamic gain at high frequencies. The main effect of increasing sodium peak conductance in the model was a substantial lowering of the current and voltage

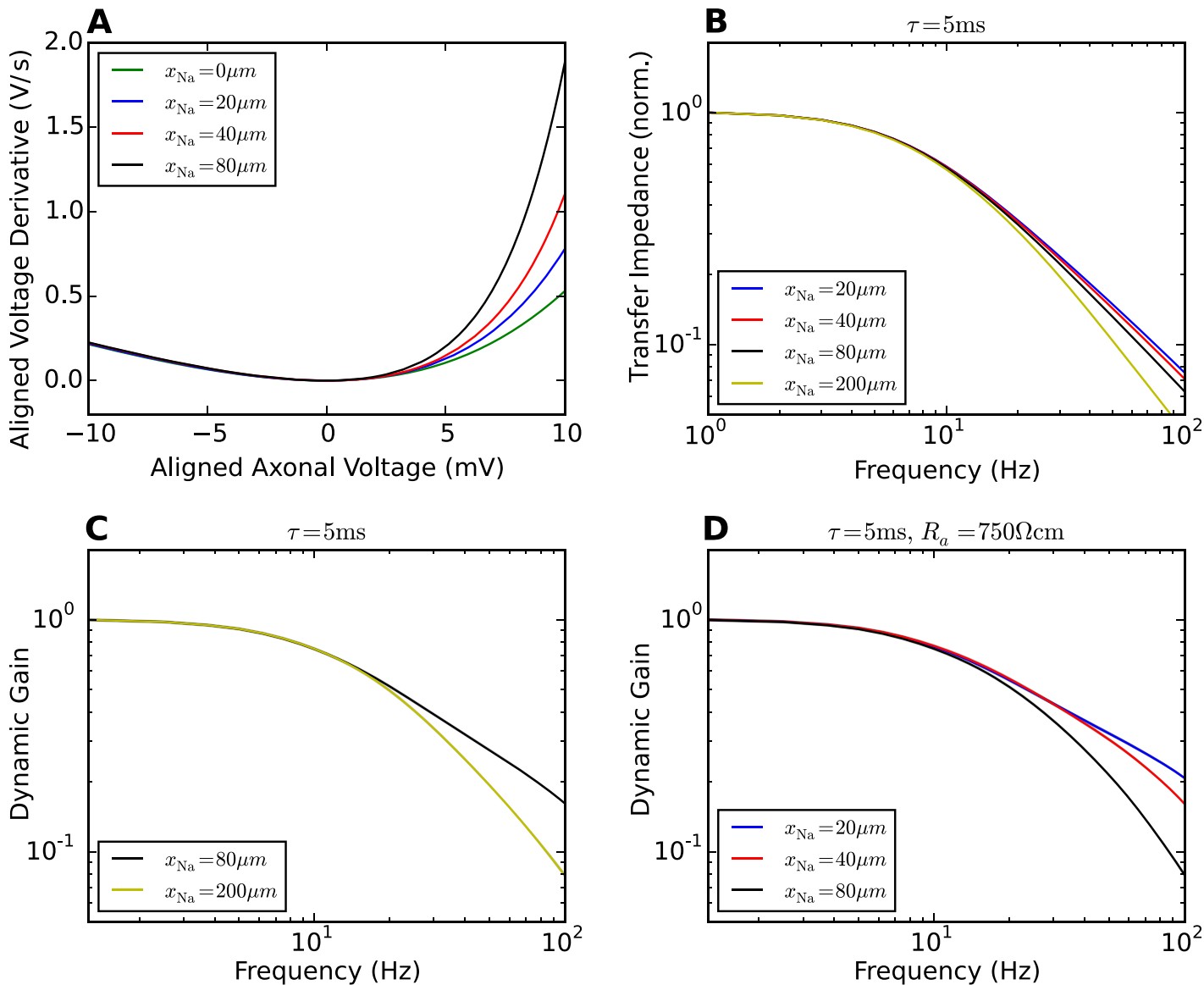

**Fig 3. Impact of initiation site distance and transfer impedance on dynamic gain. A** Phase plots for the AP waveforms at the initiation site. Moving the initiation site away from the soma reduces the lateral current during AP generation, leading to a more rapid initiation dynamics. The local minima of all curves are aligned at (0mV, 0mV/ms). The action potential waveforms are shown in S1 Fig. **B** Transfer impedance when the stimulus is transmitted from the soma to the AP initiation site. **C** Increasing $x_{Na}$ to 200$\mu$m decreases the high-frequency gain of the neuron model. **D** Increasing $R_a$ to 750$\Omega$cm reduces the impact of initiation site distance on dynamic gain (compare to Fig 2B). The transfer impedance effect dominates the dynamic gain. A larger $x_{Na}$ reduces the high-frequency gain.

threshold of the model. However, static gain around threshold, i.e. the slope of the F-I curves, was largely preserved. The inset in panel **C** shows the F-I curves for the three sodium peak conductances used. Note that the F-I curves are shifted along the current axis. This essentially preserves gain as measured by the curve's slope at a fixed firing rate.

Fig 4B and 4D show the phase plots of AP initiation dynamics for constant stimuli that reproduce 1Hz and 5Hz firing rate separately. In both conditions, increasing the sodium peak conductance by a factor of 5 accelerated AP initiation. A further doubling had less impact in the voltage range displayed here. Comparing the AP initiation dynamics with the dynamic

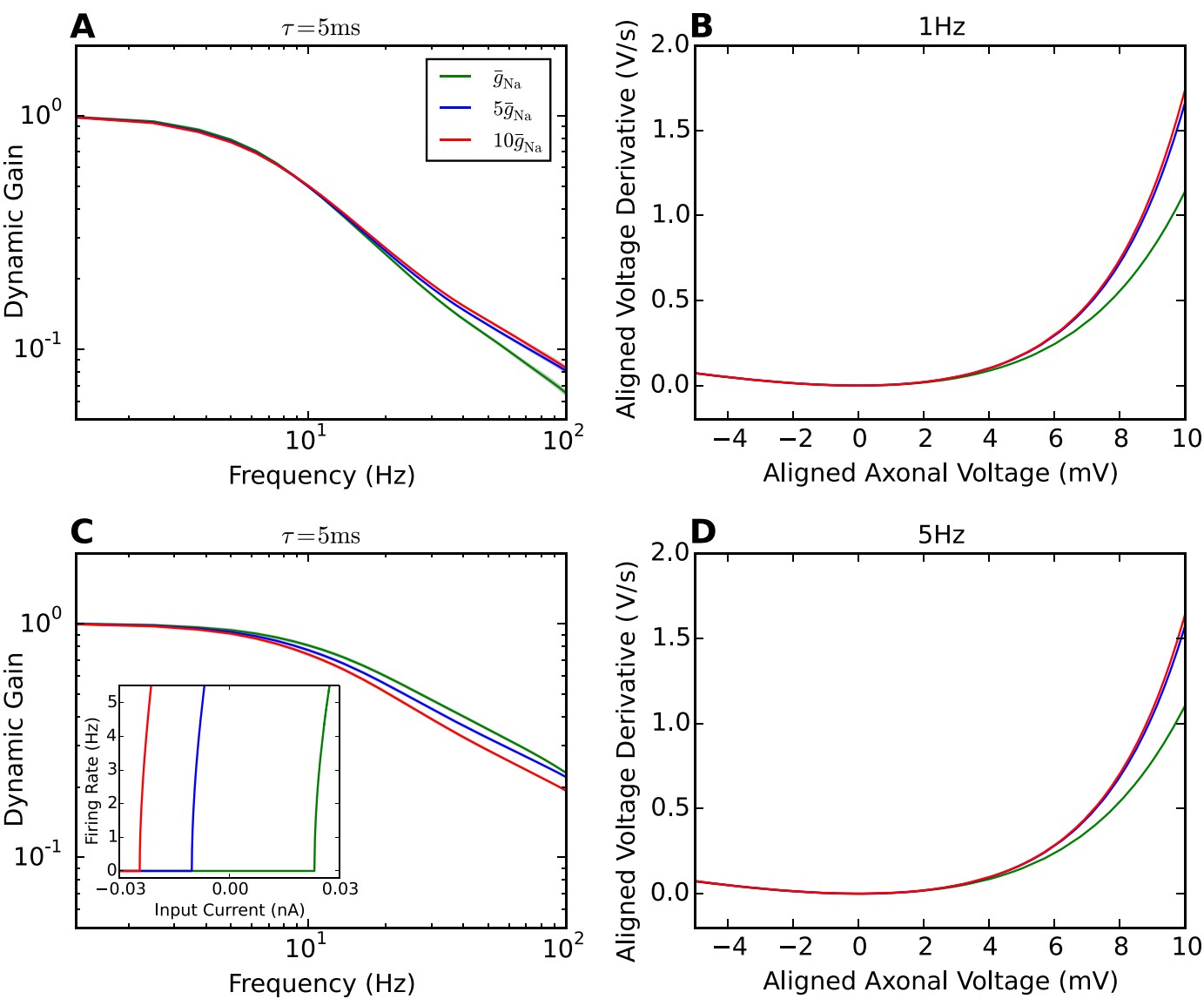

**Fig 4. AP initiation dynamics and dynamic gain comparisons for difference sodium peak conductances. A** Dynamic gain functions of neuron models with different sodium peak conductances for 1Hz firing. The CV is 0.9±0.05. **B** Phase plots of AP initiation dynamics of Brette's models with different sodium peak conductances. $x_{Na}$ is fixed to 40$\mu$m. Sodium peak conductance $\bar{g}_{Na}$ is increased 5 and 10 fold separately in comparison with the original model. To compare the AP initiation speed, each neuron model is injected with a constant input that generates 1Hz firing rate. **C** Dynamic gain functions of neuron models with different sodium peak conductances for 5Hz firing. The CV is 0.85±0.05. The inset in panel **C** shows the F-I curve for the three sodium peak conductance used. **D** Phase plots of AP initiation dynamics at 5Hz firing.

gain functions, at 1Hz firing rate, the dynamic gain enhancement parallels the AP upstroke acceleration for increasing sodium peak conductance. For the 5Hz firing rate, the small changes in dynamic gain at 100 Hz appear dissociated from the changes of AP dynamics.

We next combined the variation of initiation site location and sodium peak conductance. The resulting cutoff frequencies of the dynamic gain are displayed in Fig 5 at fixed firing rates of 5Hz and 1Hz. The exact working points were found by fixing the CVs of the ISI distributions to 0.9 ± 0.05 for Fig 5A and 5B, or by fixing the std of stimuli for Fig 5C and 5D. The 2D surface plots show the parameter dependence of the cutoff frequencies and the decay of

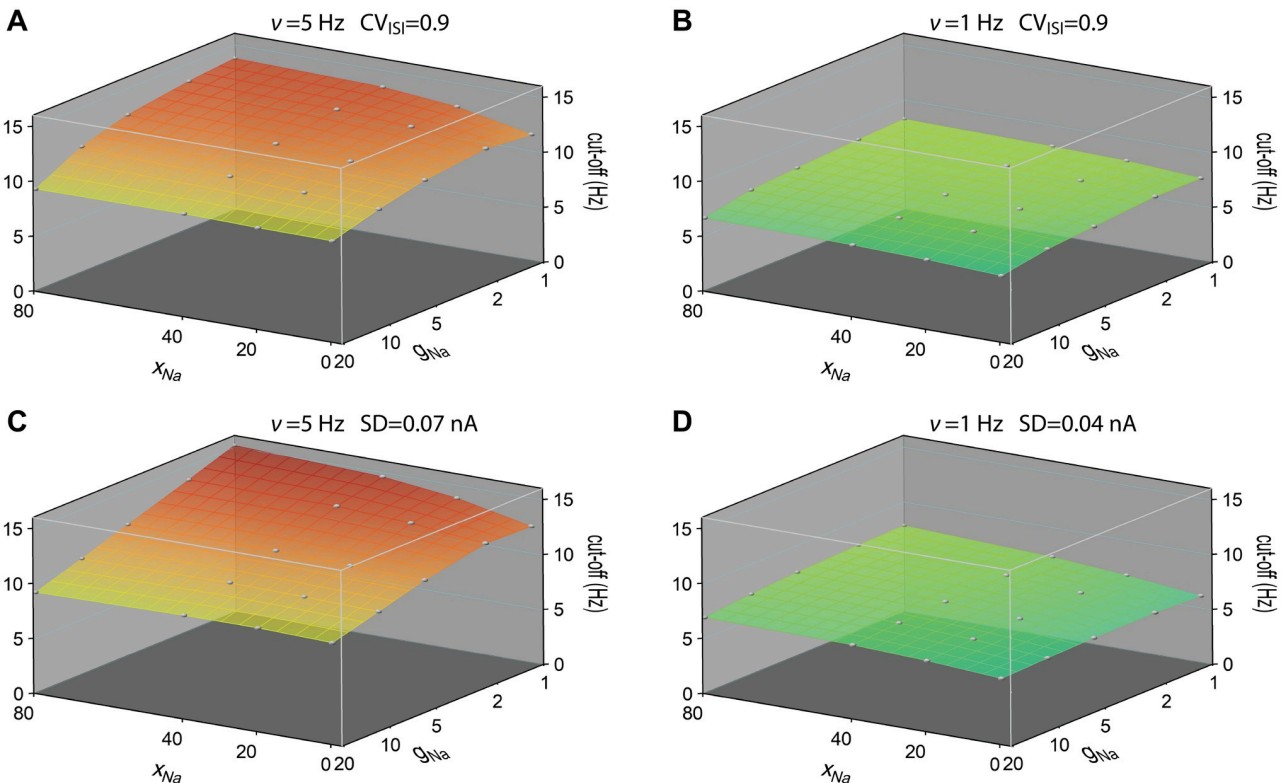

**Fig 5. Cutoff frequencies of dynamic gain curves as a function of sodium peak conductance and AP initiation site. A** and **B** For each pair of AP initiation site and sodium peak conductance, the dynamic gain curve was calculated at 5Hz and 1Hz firing rates working points. CV values were fixed at $0.9 \pm 0.05$. **C** and **D** The dynamic gain curves were calculated at 5Hz and 1Hz firing rate with std of stimuli fixed at 0.07nA and 0.04nA. Cutoff frequency was the frequency to which the dynamic gain decayed by a factor of $\sqrt{2}$. Correlation time of background current was 5ms. $x_{Na}$ ranged from $0\mu m$ to $80\mu m$. Sodium peak conductance ranged from $1\bar{g}_{Na}$ to $20\bar{g}_{Na}$, represented on a logarithmic scale. Grey points represent calculated data, continuous surfaces are obtained by Voronoi interpolation to allow for a continuous color representation.

dynamic gain values in the low frequency region. At 5Hz firing rate, both stimulus conditions in **A** and **C** show that the cutoff frequency is higher with larger $x_{Na}$ and smaller sodium peak conductance. At high sodium peak conductance, the cutoff frequency is less sensitive to the AP initiation site. At 1Hz firing rate, we observed a similar trend of the cutoff frequency. Compared to the high firing rate cases, at 1Hz firing rate, the 2D surface plots are flatter for both stimulus conditions. Besides, the CV of the ISI distribution was less dependent on the mean at 1Hz firing rate. Although the dynamic gain values in the high-frequency region are enhanced for higher sodium peak conductance at 1Hz firing rate, the cutoff frequencies are slightly lower for high sodium peak conductance models. When fixing the std, we obtained a similar dependence.

## Weak high-frequency boost when axonal currents show moderate voltage sensitivity

Next, we examined the impact of correlation time of background current fluctuations on the dynamic gain curve (Fig 6). We increased it from $\tau$ = 5ms to 50ms and calculated the dynamic gain functions for $x_{Na}$ = 20$\mu$m, 40$\mu$m, 80$\mu$m. Distinct from the behaviour of LIF-like models [17, 24, 25, 35, 36] and cortical neurons [7], increasing the correlation time did not enhance

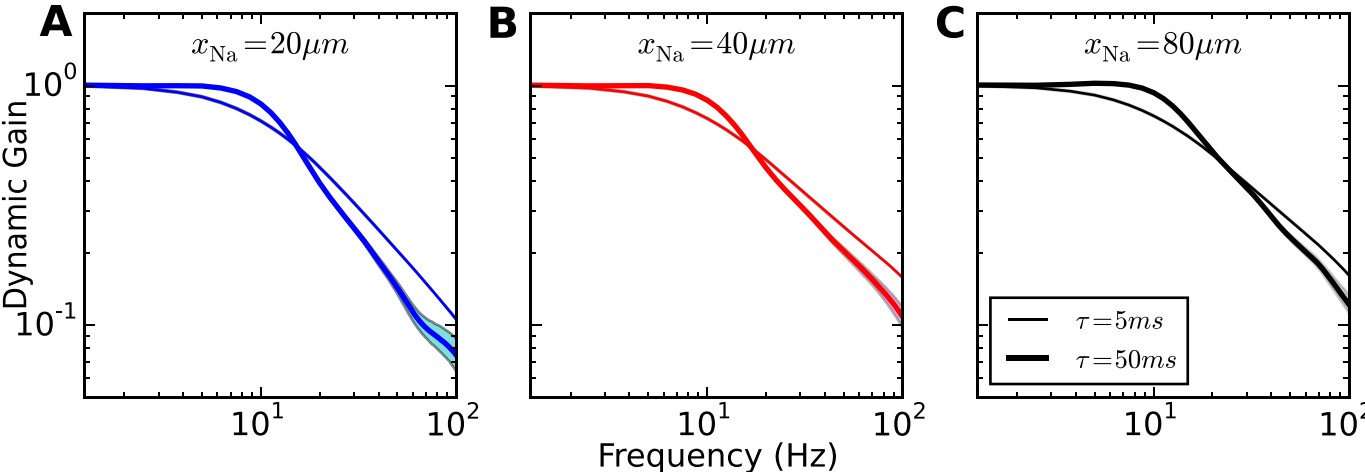

**Fig 6. Impact of background current correlation time on dynamic gain.** Dynamic gain functions for different input current correlation times. Model properties as for Fig 2. $x_{Na}$ increases from = 20, 40 to 80$\mu$m (**A**, **B**, **C**). Line width indicates input correlation time (thin: $\tau$ = 5ms, wide: $\tau$ = 50ms).

the dynamic gain in the high-frequency range. We observed a weak resonance around 10Hz, i.e. a slight enhancement of dynamic gain below the cutoff frequency. For higher frequencies, the dynamic gain decayed steeper than that for $\tau$ = 5ms. This observation holds for all three $x_{Na}$ values, and shows that overall the dynamic gain of the model is quite insensitive to input current correlation time. Independent of the input correlation time, this model's dynamic gain had a much lower bandwidth than any of the experimentally characterized cortical neurons, which exhibit cutoff frequencies in the range of hundreds of Hz [3–15, 37, 38]. The insensitivity of the dynamic gain to the current correlation time qualitatively deviates from experimental observations [7]. In summary, our results show that shifting the position of the AP initiation site away from the soma does increase the onset rapidness of APs observed at the soma, but sharp somatic APs onset does not by itself induce an ultrafast population response.

## Increasing axonal current's voltage sensitivity establishes ultrafast population encoding

The results above suggest that the AP voltage dynamics at the AP initiation site is more closely related to the high-frequency encoding properties than the somatic AP waveform. We first attempted to accelerate this dynamics by reducing the capacitive load in the axon. But even when we assumed the axonal segment beyond the initiation site to be myelinated, the bandwidth did not increase (S4 Fig). We thus turned to more direct methods and examined whether increasing the voltage sensitivity of the sodium current at the initiation site could enhance high-frequency gain and potentially equip the multi-compartment model with an ultrafast population response.

To test this, we changed the parameter $k_a$ of the activation function of the sodium gating variable from 6mV to 0.1mV making the sodium current activation curve nearly step-like. With this modification, the voltage decoupling between soma and axon was nearly instantaneous for all $x_{Na}$ tested (Fig 7A). With high voltage sensitivity, the axonal voltage and somatic voltage are nearly identical until $V_{1/2}$ is reached. Above this threshold value, a large sodium current is activated, causing an immediate substantial decoupling. Compared to the voltage decoupling with a standard sodium activation function, there was a substantial change of the decoupling trajectory for $x_{Na}$ = 20$\mu$m. However, for large $x_{Na}$ values, the decoupling

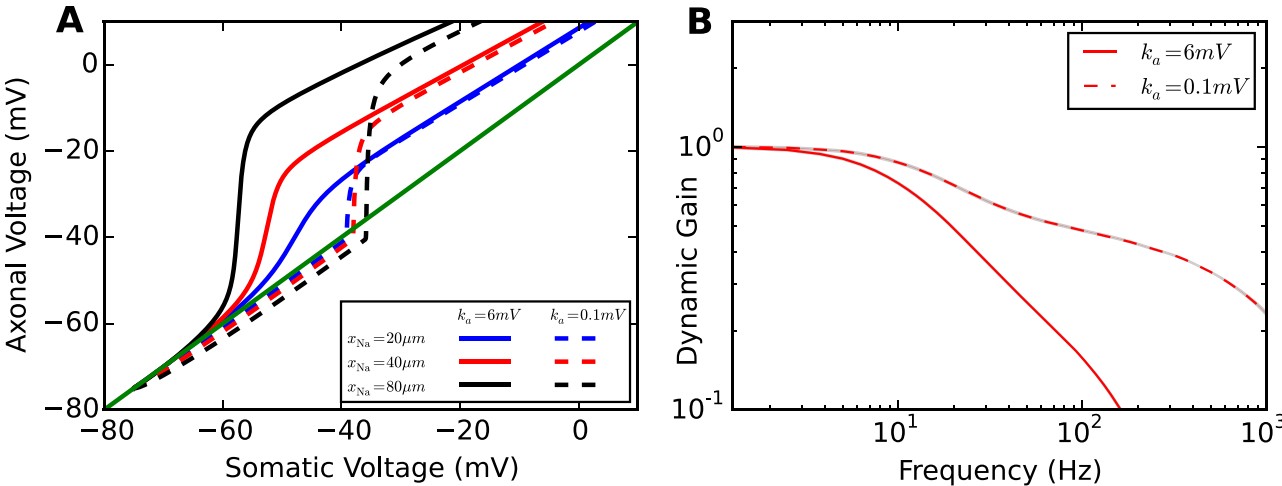

**Fig 7. Impact of voltage sensitivity of sodium current activation curve on voltage decoupling under current clamp and dynamic gain. A** Plot of axonal voltage at $x_{Na}$ vs. somatic voltage for different sodium channel positions. Solid lines are voltage traces for low voltage sensitivity $k_a = 6$mV, dashed lines denote the traces for high voltage sensitivity $k_a = 0.1$mV. **B** Dynamic gain functions for the different voltage sensitivities for $x_{Na} = 40\mu$m and $\tau = 5$ms. The dynamic gain curve for the intermediate value $k_a = 1$mV and longer correlation times can be found in S3 Fig.

trajectories are quite parallel with those for a less voltage sensitive sodium activation function. In Fig 7B, we compare the dynamic gain functions of original model to the model with a more voltage sensitive sodium activation function. We fixed $x_{Na}$ to $40\mu$m and $\tau$ to 5ms. Fig 7B shows that the dynamic gain decreases less for high frequencies and hence the bandwidth increases.

We also examined the impact of the positioning of the AP initiation site and the input current correlation time on dynamic gain for the case of high voltage sensitivity (Fig 8). Fig 8A shows that with $k_a = 0.1$mV, the high-frequency dynamic gain is greatly enhanced for all three $x_{Na}$ values. One additional observation is that increasing $x_{Na}$ decreases the dynamic gain in the high-frequency regime. This behavior is opposite to the behavior for low voltage sensitivity (Fig 2B). This phenomenon may be related to the impact of transfer impedance and lateral

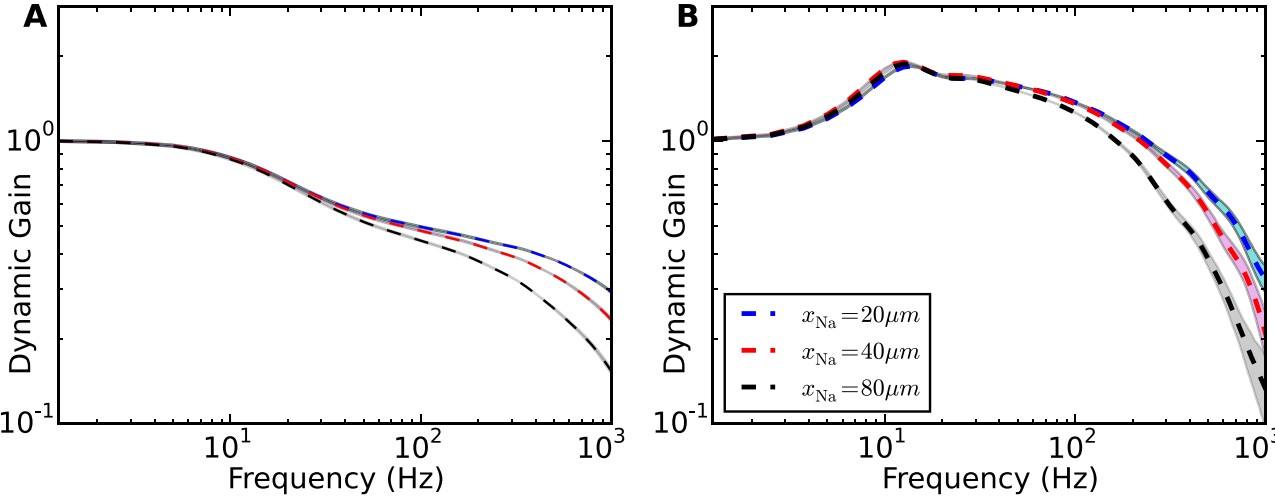

**Fig 8. Increased impact of current correlation time for high voltage sensitivity.** Dynamic gain curves are plotted for **A** a short input current correlation time $\tau = 5$ms and **B** a long correlation time of $\tau = 50$ms. Dashed lines as in Fig 7B. Shaded areas are 95% confidence intervals. Colors denote different initiation site positions (see legend).

current proposed above. When the sodium activation function is highly voltage sensitive, the lateral current has limited impact on the voltage dynamics at the initiation site. However, the transfer impedance from soma to axon still impacts the dynamic gain function in the high-frequency range. Fig 8B displays the dynamic gain functions for $\tau$ = 50ms. For higher voltage sensitivity, increasing the input current correlation time increases high-frequency gain similar to the behavior first described for the LIF neuron [17]. A more extensive overview of the combined effects of correlation time, axonal channel position and voltage sensitivity is given in S2 Fig.

The activation dynamics of sodium current during AP initiation is determined by both the voltage sensitivity of sodium activation curve $k_a$ and the sodium peak conductance $\bar{g}_{\mathrm{Na}}$. In Fig 9, we display the cutoff frequency as a function of these two parameters. We fixed $x_{\mathrm{Na}}$ at 40$\mu$m and correlation time of background current $\tau$ at 50ms. $k_a$ ranged from 0.1 to 6mV. $\bar{g}_{\mathrm{Na}}$ was increased from 1 fold to 20 fold. We fixed the firing rate and CV (Fig 9A and 9B), or firing rate and std of stimulus (Fig 9C and 9D). For all four conditions, increasing the voltage sensitivity of sodium activation curve increased the cutoff frequency in agreement with previous theoretical studies [17, 24, 28, 29]. The cutoff frequencies for 5Hz firing rate were in general larger than those for 1Hz firing rate at given $k_a$ and $\bar{g}_{\mathrm{Na}}$. Overall, the cutoff frequencies were much less sensitive to $\bar{g}_{\mathrm{Na}}$ than to $k_a$. Fig 5 shows that at $k_a$ = 6mV and $\tau$ = 5ms, the cutoff frequency is slightly decreased when increasing the sodium peak conductance. For $k_a$ = 6mV and $\tau$ =

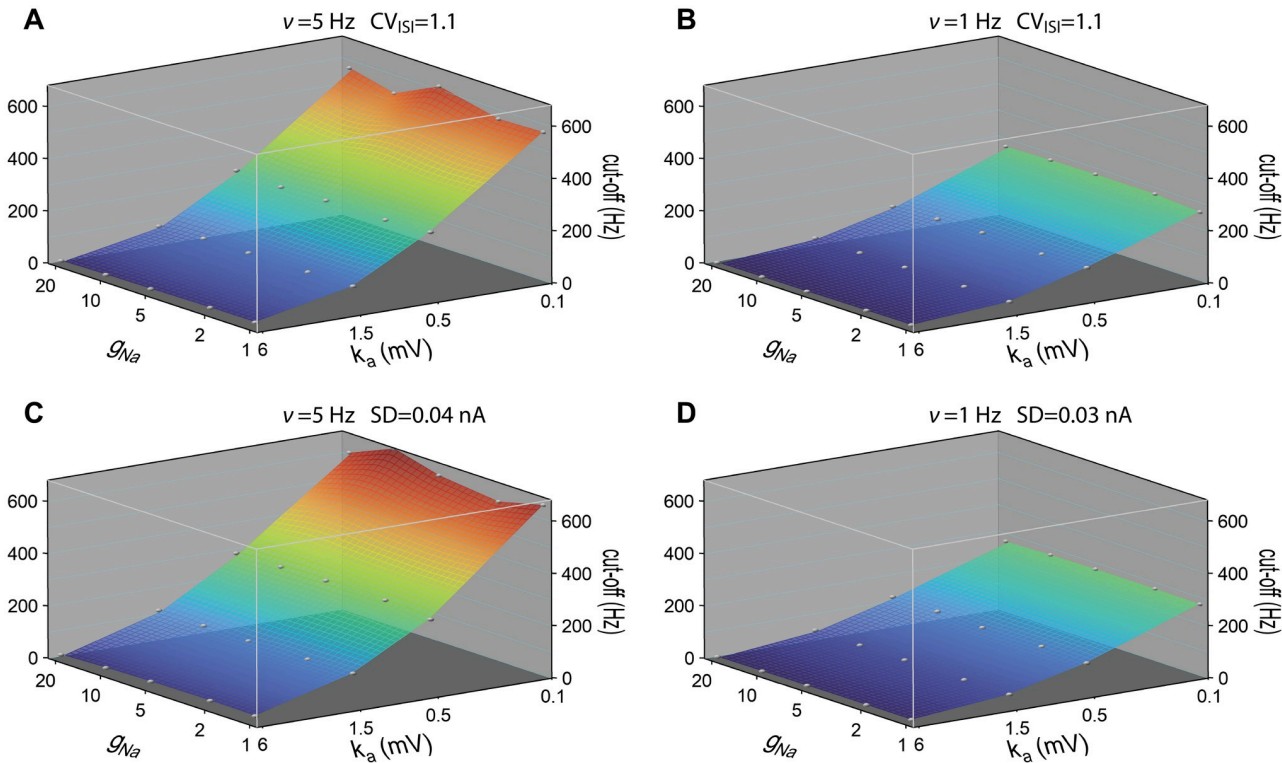

**Fig 9. Cutoff frequencies of dynamic gain curves as a function of sodium peak conductance and voltage sensitivity of sodium activation curve. A** and **B** For each pair of voltage sensitivity and sodium peak conductance, the dynamic gain curve was calculated at 5Hz and 1Hz firing rates working points. CV values were fixed at 1.1 ± 0.05. **C** and **D** The dynamic gain curves were calculated at 5Hz and 1Hz firing rate with std of stimuli fixed at 0.04nA and 0.03nA. Correlation time of background current was 50ms. $x_{\mathrm{Na}}$ fixed at 40$\mu$m. Sodium peak conductance ranged from $1\bar{g}_{\mathrm{Na}}$ to $20\bar{g}_{\mathrm{Na}}$. $k_a$ ranged from 0.1 to 6mV. Conductance and sensitivity are represented on a logarithmic scale. Grey points represent calculated data, continuous surfaces are obtained by Voronoi interpolation to allow for a continuous color representation.

50ms, a higher sodium peak conductance also led to a slightly lower cutoff frequency. For other values of $k_a$, the cutoff frequency remained relatively insensitive to the sodium peak conductance.

### Excessive soma size improves high-frequency dynamic gain

So far, we have not found a condition for which resistive coupling substantially impacts the cutoff frequency. We argued that the bifurcation observed for a voltage-clamped soma does not manifest during dynamic AP initiation in the unclamped neuron (Fig 6 continuous vs. dashed line). But there is a possibility to render the somatic voltage almost insensitive to lateral currents, even though it is not actively voltage-clamped: the soma size has to be increased and thereby, the somatic capacitance. We next study the impact of soma size on the dynamic gain.

In the following simulations, $d_S$ and $l_S$ ranged from a physiologically plausible size of $10\mu m$ up to 2cm. $x_{Na}$ was $40\mu m$, and the input current correlation time was 5ms. For the 2cm case, the spatial grid of the soma was increased to $1000\mu m$ for the convenience of simulation. As depicted in Fig 10A, the colored thick lines are the unnormalized dynamic gain curves for various soma sizes. To generate the same firing rate in these models, similar amounts of current reached the AP initiation site. However, to overcome the leak current in the soma and generate somatic voltage fluctuations, the somatic input had to be scaled with the surface area of the soma. As a result, the dynamic gain amplitudes were greatly attenuated for the larger soma cases.

Although of "enormous" length in the most extreme case, the somatic compartment was still electrotonically compact in these simulations. To test whether the transfer impedance was in fact minimal, we kept the soma surface areas and reduced the soma length to $2\mu m$ and recalculated the corresponding dynamic gain curves. The black lines in Fig 10A demonstrates that transfer impedance in the soma has negligible influence on the population dynamic gain.

In Fig 10B, we compared the normalized dynamic gain curves for different soma sizes. Increasing the soma size enhanced the high-frequency dynamic gain. However, the dynamic gain curves did not become flat as expected for a LIF-like model. As a function of soma size,

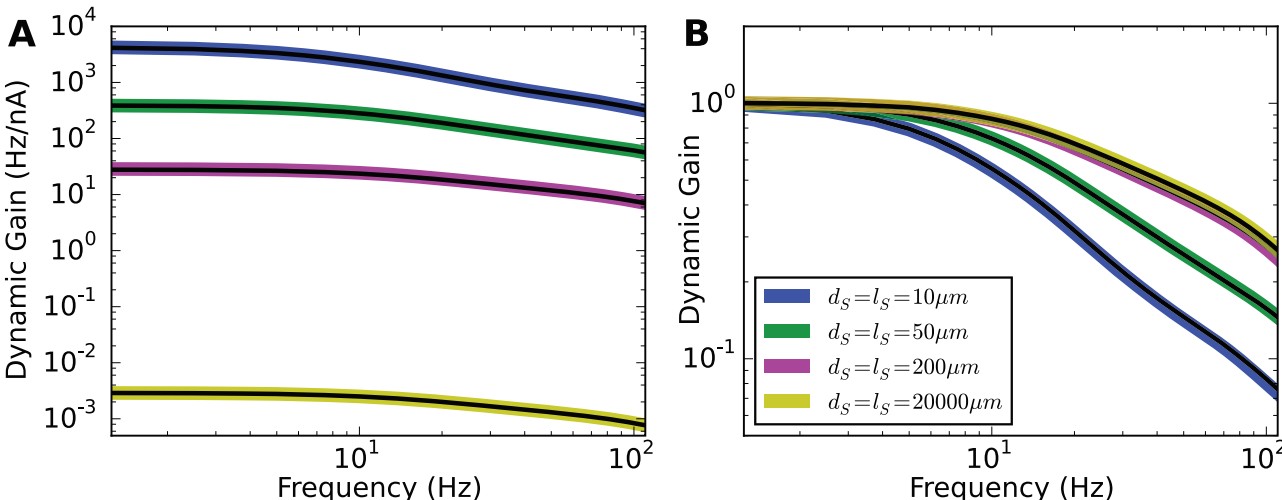

**Fig 10. Effect of soma size on dynamic gain. A** Unnormalized dynamic gain curves for various soma sizes. Diameters and lengths for all the soma are shown in the legend panel in **B**. $x_{Na}$ was fixed to $40\mu m$. The correlation time $\tau$ was fixed to 5ms. For each soma size, fixing surface area but reducing soma length to $2\mu m$, the dynamic gain curves were recalculated and represented as black lines, which overlapped with corresponding colored lines. **B** Comparison of all the normalized dynamic gain curves. Increasing the soma size enhances the dynamic gain in the high-frequency regime for Brette's original model.

the normalized dynamic gain seemed to converge to a limiting curve, with little change beyond soma sizes of 200$\mu$m. The deviation of the neuron model from the LIF model may be explained by the position of the AP initiation site. When the sodium channels are positioned at the soma, for a given sodium peak conductance, increasing the soma size decreases the ability of sodium current to change the voltage derivative. In the large soma size limit, the sodium current would not suffice to overcome the leak current for AP generation, so that the neuron model will be closely approximated by a model with hard threshold driven only by the external input, i.e. a LIF model. However, moving the AP initiation site away from the soma along the axon reduces the upper limit of the lateral current shaping the AP initiation dynamics. As for our case where $x_{\mathrm{Na}}$ is set to 40$\mu$m, some upstroke dynamics was retained even when the soma was extremely large. The population encoding ability in the large soma model limit seems bounded by the sodium activation dynamics at the initiation site.

In Fig 11, we analyzed the transfer impedance and AP initiation dynamics of Brette's models with different soma sizes. As a comparison, we also present the analysis for Brette's models with high voltage sensitivity of the sodium activation function. We will show that the enhanced encoding ability seen in Fig 10 originates from the acceleration of axonal AP initiation dynamics.

An increase in soma size changes the input encoding in two ways. The filter from input current to voltage at the initiation site is changing and an increased lateral current drains more of the depolarizing sodium current. The transfer impedance, shown in Fig 11D, decays faster with a bigger soma. This effect dominates for the steep sodium activation curve ($k_a$ = 0.1mV) and underlies the reduction in dynamic gain in Fig 11B. The second effect, the increased current sink of the larger soma, indeed led to a decreased AP initiation speed, but while this was almost negligible for $k_a$ = 0.1mV, it was very large for the conventional sodium activation curve ($k_a$ = 6mV) in Fig 11C. As a result, the voltage at which sodium currents overcome current drainage by leak and lateral current is shifted by more than 10mV to a region where the sodium activation curve is steeper. For better comparison, Fig 11E aligns the local minima of two axonal AP initiation dynamics. The AP initiation dynamics are slower and smoother for the small soma. In this sense, the enormous soma size accelerates the initial AP initiation, creating a steeper axonal AP waveform and thereby an enhanced dynamic gain in the high-frequency region.

## Discussion

Our study demonstrates that the experimentally observed high bandwidth encoding of cortical neurons is critically influenced by the properties of the ion channels in the AIS, and further modulated by the electrotonic structure of axo-somatic and somato-dendritic compartments. We used the dynamic gain function for an extensive comparison of model variants, designed to test the resistive coupling hypothesis [31, 32] and the impact of AIS ion channels. By choosing the same, well-defined working points for all models, we eliminated free parameters and assured unbiased comparison. Resistive coupling [39], also called compartmentalization [31], is not sufficient to account for the ultrafast population coding. We found the dynamic gain function to be weakly influenced by the separation between AP initiation site and soma, as was the initial slope of the AP phase plot at the initiation site. A weak impact on AP initiation and population encoding was also observed when we changed the sodium conductance amplitude or the soma size. Even extreme choices for these parameters were insufficient to induce a high bandwidth of the dynamic gain function. An increased voltage sensitivity of sodium channel activation, however, substantially enhanced AP initiation speed and population encoding ability. Our results indicate that the key determinant of population encoding ability is the effective

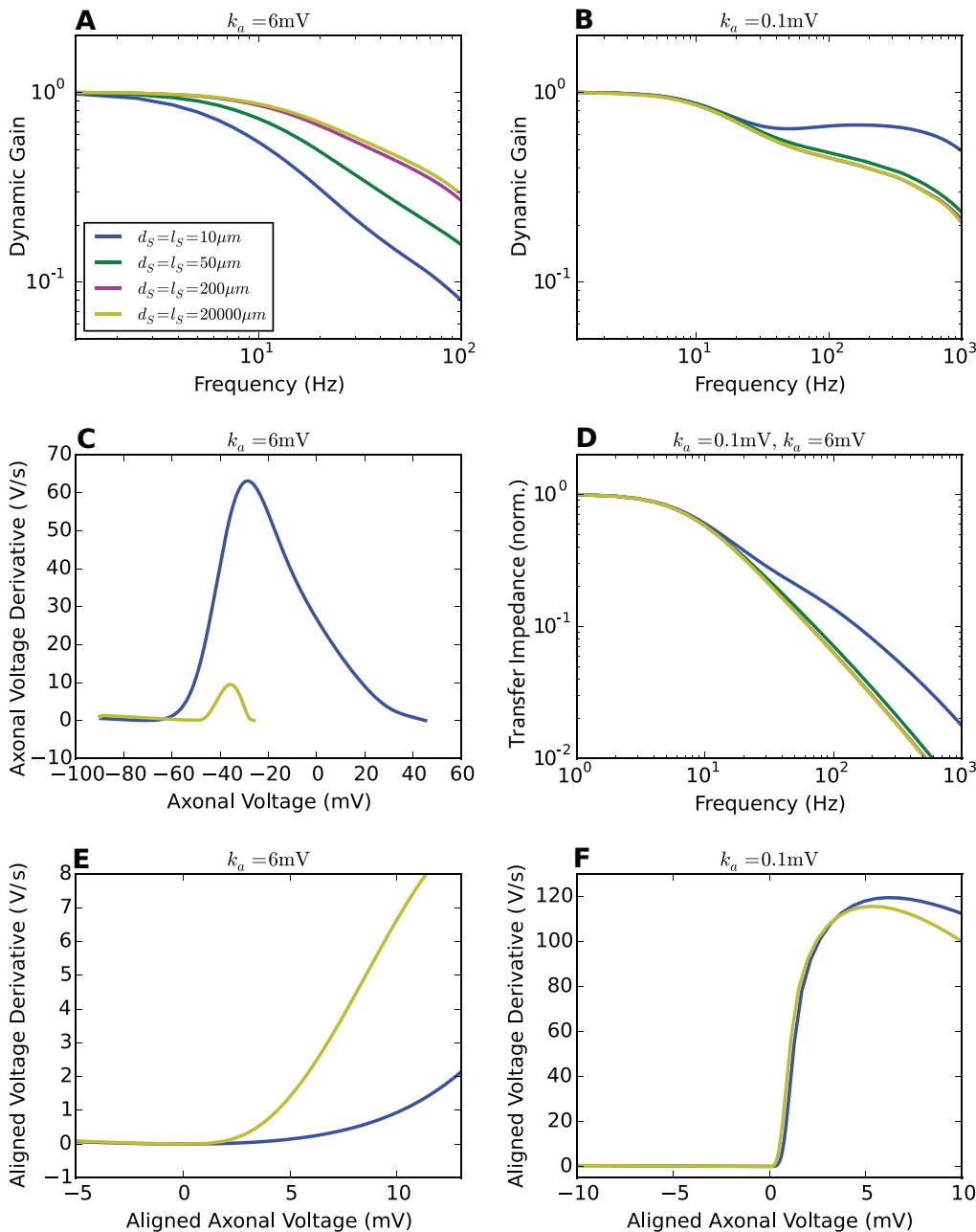

**Fig 11. Transfer impedance and AP initiation dynamics for different soma sizes. A** and **B** Dynamic gain functions of Brette's model for a variety of soma sizes. $k_a$ is set to 6mV and 0.1mV respectively. $x_{Na} = 40\mu$m. $\tau$ = 5ms. Legend panel in **A**. **C** AP initiation dynamics seen at the axon with $k_a$ = 6mV. **D** Transfer impedance for different soma sizes. The passive neuron models are identical for both $k_a$ values. **E** and **F** Axonal AP initiation dynamics with local minima aligned at (0mV, 0mV/ms).

voltage dependence of AP initiating ion current, subsumed here into a single sodium channel type. It remains an intriguing discrepancy that experimental characterizations of neuronal sodium channels report voltage sensitivities corresponding to $k_a$ = 4–6 mV [40].

The models we used for these simulations are based on a design by Romain Brette [31]. While it features fewer ion channel types than realistic models of cortical neurons, it retains

the essential features to study AP initiation dynamics in the context of neuron morphology. The model lacks repolarizing current and sodium current inactivation. As a consequence, a forcible voltage reset is required to terminate an AP. This can have a minor impact on the inter-spike interval distribution but will not affect the supra-threshold dynamics, which influences the encoding bandwidth. We drive the model by current injections rather than the changing synaptic conductances that underlie fluctuating input *in vivo*. The range of working points that can be realized with this paradigm might be even more extensive than is physiologically accessible. Therefore our findings should also hold within the physiological range of working points.

The close relation between the voltage dependence of AP initiation currents and the encoding bandwidth is consistent with previous theoretical studies. For model currents with exponential [24] or piecewise linear voltage dependence [28], increases in the voltage dependence substantially improved the encoding of high-frequency input components. The low cutoff frequency that we report here for the model from [31] with standard voltage sensitivity resembles the results of single compartment conductance-based models [24, 29] in which a similar voltage dependence was used. Encoding of neurons with an infinite voltage sensitivity, represented by the fixed AP threshold in integrate-and-fire models, was first considered by [16] and more extensively studied by [17]. We closely reproduce its encoding behavior when we increase the axonal current's voltage sensitivity. A remaining difference is the finite bandwidth, plausibly caused by the finite activation speed of our model, $\tau_{act}$ = 0.1ms as compared to the instantaneous activation in a LIF model.

We report a modulatory impact of neuron morphology, which is consistent with previous studies [11, 33, 41]. The neuron's electrotonic structure affects the effective impedance transforming fluctuating current input into voltage at the initiation site. [11, 33, 41] showed that for current injections proximal to the initiation site, the impedance of a large somato-dendritic compartment causes an apparent boosting of high-frequency components. On the other hand, an increasing distance between injection site and initiation site can attenuate high-frequency components, as we show in Figs 3 and 8. For simple morphologies, such passive effects can be understood analytically [42]. Our analysis reveals another effect of the electrotonic structure, mediated by an altered AP initiation dynamics. An increased somatic current sink drastically reduces the depolarizing currents available at the initiation site. However, the voltage threshold for AP initiation is also strongly shifted towards the voltage at which the ion currents attain their peak voltage dependence (Fig 11). These two effects have opposing impact on the activation dynamics at threshold and consequentially, the combined effects of increased somato-dendritic size can reduce the bandwidth, as in the case of high voltage sensitivity $k_a$ = 0.1mV, or increased the bandwidth, as in the case of a low voltage sensitivity $k_a$ = 6mV, similar to the AP waveform change reported by [33]. While several of our findings have been reported before in different models and with different emphasis, our study offers the first exhaustive comparison of models, preserving an invariant firing rate and ISI distribution under all parameter variations. It thus provides an unambiguous dissection of the different passive and active effects.

Over the past decade, the simple picture of the initial axon as a high channel density focus in an otherwise uniform axonal membrane has been superseded by findings of an ever more intricate nanoscale organisation of the AIS. It is structured by the interplay of cell-adhesion, cytoskeletal and scaffolding proteins and nanoscale ion channel clusters [43–46]. Dynamic gain studies consistently suggest that the AIS molecular nano-architecture is under substantial selective pressure to equip the population of hundreds to thousands of cells to which a particular neuron belongs, with a high encoding bandwidth [9, 14, 18]. From this perspective, the AIS appears as an intricately structured service organelle that is built and maintained by the individual neuron but fulfills its essential function only at the higher level of populations and

neuronal circuits. The population coding speed hinges on the voltage sensitivity of the AP initiation current, and is modulated by the electrotonic structure of the neuron. In real neurons, more than one ion channel type contributes to the AP initiation dynamics and a pioneering study has already shown that not only Na currents, but also Kv1 potassium currents, activating around AP threshold, improved AP locking to high input frequencies [6]. The critical importance of axonal conductances is further supported by observations of reduced encoding bandwidth following local manipulations of axonal conductances through pharmacological intervention or mutation of the axon cytoskeleton [9, 14].

Systematic studies connecting AIS molecular architecture, to theory-driven, statistical analysis of population encoding performance are required to understand the impact of AIS molecular design, plasticity and pathology on population level information processing. Dynamic gain measurements are currently the most sensitive tool for such an approach, as demonstrated by recent experimental and theoretical studies [14, 41]. However, it is crucial that the comparison across neurons is performed systematically and without bias. Specifically, the AP trains evoked in such experiments should have the same overall statistics, because the dynamic gain function of a neuron depends on the operating point it is tested at. Here, we fixed the firing rate and the CV of the ISI distribution, following [47]. Such standardization assures that the firing patterns of different simulations have similar entropy, highlighting the actual population encoding differences across models.

In our dynamic gain analysis, we observed a clear impact of the voltage dependence of the axonal currents on the AP initiation dynamics and the bandwidth of population encoding. Several experimental studies have shown that the AIS can undergo plastic changes when their activity levels are changed [48–54]. In the auditory system, deprivation of external input not only increased the length of AIS and sodium channel densities [49] but also changed the dominant potassium channel type [50], which enhanced the neuronal excitability. In response to increased stimulation, however, excitatory neurons in different systems were found to become less excitable and to shift their axonal ion channels away from the soma by several micrometers [51, 53, 54]. In the simple model that we studied here, such smalls shifts did not lead to a pronounced change in dynamic gain or excitability, however, the study by Kuba et al. [50] raises the intriguing possibility that AIS plasticity also affects the composition of the axonal ion channels and thereby the dynamic gain [14]. Post-translational modifications of ion channels through enzymes or modulation through auxiliary subunits are expected to impact dynamic gain within minutes or hours. Studies of AIS plasticity should therefore be complemented by measurement of the dynamic gain. A particularly drastic change of operating conditions, transient hypoxia and spreading depolarization has recently been shown to lead to a disruption of the AIS organization, and in this case the bandwidth of the dynamic gain was indeed substantially reduced [18]. While AIS plasticity can adjust neuronal excitability in a relatively short period of time, the dendrite morphology might impact the dynamic gain on developmental time scales or cause differences between species [10, 21, 23] or individuals affected by AIS pathologies [55]. Our dynamic gain analysis of Brette's model provides an approach and baseline for analyzing different neuronal populations, mutants and manipulations as well as multi-compartment models. This analysis approach can be generalized to study how neuron morphology and axonal conductances contribute to the dynamic gain in more complicated scenarios.

## Methods

### Model

Simulations were performed with NEURON versions 7.3, 7.6 and 8.0 [56] as a module for Python 2.6, 3.7, or 3.9. All code is available at https://github.com/chenfeizhang/Brette_gwdg.

We used a ball-and-stick neuron model composed of a soma and an axon. All parameters were identical to the model in [31], the equivalence of the models was confirmed by reproduction of published results (see Fig 1A). Here, the somatic compartment was modeled as a cylinder with equal length and diameter, which has the same effective capacitance as the sperical soma in [31]. The soma diameter was $d_S = 50\mu m$ and its length was $l_S = 50\mu m$. The axon had diameter $d_A = 1\mu m$ and length $l_A = 600\mu m$. Passive properties of the soma and axon are listed below. Axial resistivity was $R_a = 150\Omega cm$; specific membrane capacitance was $c_m = 0.75\mu F/cm^2$ and specific membrane resistance was $R_m = 30000\Omega cm^2$ with leak reversal potential of $E_L = -75mV$. The only voltage dependent conductance in the model represents sodium current at a single location, at a distance $x_{Na}$ away from the soma. This was modeled as a NEURON point process. This sodium current has a Boltzmann activation curve similar to experimental observations [57] ($V_{1/2} = -40mV$ $k_a = 6mV$). This was combined with a voltage independent activation and deactivation time constant of $\tau_m = 0.1ms$. As in [31], sodium channel inactivation was ignored. The sodium peak conductance was $\bar{g}_{Na} = 5.23 \cdot 10^{-9}S$. The reversal potential of the sodium current was $E_{Na} = 60mV$. In a subset of simulations, the voltage dependence of the sodium current was changed as noted in the results.

The neuron model was thus defined by the following system of differential equations.

$$c_m \frac{\partial V_S(x,t)}{\partial t} = \frac{d_S}{4R_a} \frac{\partial^2 V_S(x,t)}{\partial x^2} - \frac{V_S(x,t) - E_L}{R_m} + \frac{I(t)}{\pi d_S} \delta\left(x - \frac{l_S}{2}\right) \tag{1}$$

$$c_m \frac{\partial V_A(y,t)}{\partial t} = \frac{d_A}{4R_a} \frac{\partial^2 V_A(y,t)}{\partial y^2} - \frac{V_A(y,t) - E_L}{R_m} - \frac{\bar{g}_{Na} m (V_A(y,t) - E_{Na})}{\pi d_A} \delta(y - x_{Na}) \tag{2}$$

where

$$\tau_m \dot{m} = m_\infty(V_A) - m$$

and

$$m_\infty(V_A) = 1/(1 + \exp((V_{1/2} - V_A)/k_a)).$$

The voltage at the soma was denoted $V_S(x, t)$ with $x \in [0, l_S]$. The voltage in the axon was denoted $V_A(y, t)$ with $y \in [0, l_A]$. The stimulus current was injected in the middle of the soma, and denoted $I(t)$. $\delta(\cdot)$ was the Dirac delta function.

The boundary conditions of Eqs (1) and (2) were given as $V_S(l_S, t) = V_A(0, t)$, $d_S \frac{\partial^2 V_S(x,t)}{\partial x^2}\big|_{x=l_S} = d_A \frac{\partial^2 V_A(y,t)}{\partial y^2}\big|_{y=0}, \frac{\partial V_S(x,t)}{\partial x}\big|_{x=0} = 0$, and $\frac{\partial V_A(y,t)}{\partial y}\big|_{y=l_A} = 0$. The first two conditions implied the continuity of voltage and current at the connecting point of soma and axon. The other two conditions represented the sealed ends of the neuron model.

## Simulations

To drive the neuron model, we injected Gaussian colored noise currents $I(t)$ into the soma as in prior experimental and theoretical studies [4–6, 9, 10, 35, 37]. We recorded the voltage at the position of the sodium channels in the axon and in the middle of soma. We refer to these as axonal and somatic voltage. We used the default backward Euler method in NEURON to integrate Eqs (1) and (2) with a time step of $\Delta t = 25\mu s$ and a spatial grid of $\Delta x = 1\mu m$.

An AP was detected, when the axonal voltage reached a detection threshold. For each variant of the model, this detection threshold was chosen as the axonal voltage at which the AP evolved most rapidly, i.e. the point of maximal rate of voltage rise. In this way, the AP detection time was least influenced by the ongoing random current fluctuations.

Because the neuron model does not feature sodium channel inactivation nor repolarizing voltage activated currents, we terminated each AP by a forced reset. Specifically we globally reset the membrane voltage to $E_L$ and the sodium gating variable $m$ to $m_\infty(E_L)$. The reset is performed upon crossing the reset threshold, which is chosen such that on average reset occurs 2ms after the AP detection threshold was crossed.

We used an Ornstein-Uhlenbeck (OU) process [58, 59] to create the current stimuli, which are characterized by their correlation time $\tau$, standard deviation $\sigma$ and mean current $\mu$:

$$\tau dI(t) = (\mu - I(t))dt + \sqrt{2\tau}\sigma dW(t)$$

where $W(t)$ denotes a Wiener process with zero mean and unit variance. A given set of mean and standard deviation results in AP sequences that can be characterized by the average firing rate $\nu$ and the CV of the ISI distribution. Changes in the input parameter do change the operating point $(\nu, CV_{ISI})$ of the neuron model, and thereby also its dynamic gain function. Therefore, we compared different model variants at the same operating point: $\nu = 5 \pm 0.25$Hz and $CV_{ISI} = 0.85 \pm 0.05$, representing in the fluctuation-driven regime in which cortical pyramidal neurons operate *in vivo*. For a subset of simulations, different operating points were chosen, as stated in the results. The OU parameters $\sigma$ and $\mu$ that drive the model to this operating point were determined in independent simulation runs.

For studies of the impact of sodium peak conductance on population encoding, the voltage values were reset to -90mV after an AP to maintain type 1 excitability throughout the whole range of sodium peak conductances.

To compare the AP waveforms for different model variants, we injected each model with the constant input that generated the target firing rate. The local minima of phase plots were aligned at (0mV, 0mV/ms).

## Linear response, dynamic gain and transfer impedance

To compute the dynamic gain of a neuron model, that is, its linear response function from the injected current to the firing rate, we followed [37] as outlined below. The linear response function was given by

$$L(f) = \frac{\mathcal{F}(C_{I\nu}(\tau))}{\mathcal{F}(C_{II}(\tau))} \tag{3}$$

where $\mathcal{F}$ denotes the Fourier transformation, $C_{I\nu}$ and $C_{II}$ the input-output correlation and input auto-correlation functions, respectively. The Fourier transform of the input auto-correlation equals the power spectral density of the input, according to the Wiener-Khinchin theorem. For an Ornstein-Uhlenbeck process, the power spectral density is $P(f) = \frac{2\tau\sigma^2}{1+(2\pi\tau f)^2}$.

To compute the input-output correlation function we made use of the following equality

$$C_{I\nu}(\tau) = \langle \Delta I(t+\tau)\Delta\nu(t)\rangle = \frac{1}{T}\int dt \Delta I(t+\tau)\left(\sum_i \delta(t-t_i) - \langle\nu\rangle\right) \tag{4}$$

where $\delta(t-t_i)$ is the Dirac delta function, $\Delta\nu$ and $\Delta I$ denote deviations from mean firing rate and mean input respectively. $T$ is the simulation time. Thus $C_{I\nu}(\tau)$ is the zero-average spike-triggered average current $(STA_I)$ multiplied with the firing rate $\langle\nu\rangle$. The time window to compute the $(STA_I)$ was chosen such that correlations decayed within the time window, which was fulfilled by a window of size 800ms centered about the spike time. We then computed the Fourier transform and applied a bank of Gaussian filters to the complex-valued Fourier transform components of the STA to de-noise as proposed in [37].

The dynamic gain was calculated from 20000 trials of numerical simulations of the stochastic current driven model described above. Each trial represents 20.5s of real time (with 0.5s burn-in time and 20s of actual recording). From the 20000 pairs of input current samples and output AP trains we determined the $STA_I$ through Eq (4), calculated the linear response function with Eq (3) and finally obtained the dynamic gain as its absolute value $G(f) = |L(f)|$.

For each model variant we determined a significance threshold curve according to the zero-hypothesis that the AP times are independent from the input waveform and hence the dynamic gain is zero. The zero hypothesis was realized by cyclically shifting all AP times of a given trial by the same, randomly chosen interval between 1 and 19 seconds. From all 20000 pairs of input current and shifted AP times, a dynamic gain curve was calculated. This was repeated 500 times for different random time shifts. The 95 percentile of these curves determined the significance threshold curve. Only significant parts of the dynamic gain functions are shown in the following figures. For the bootstrap confidence interval, we used the 20000 individual simulations to calculate 400 $STA_I$ curves from 50 simulations each. We then bootstrap re-sampled the grand average $STA_I$ from these 400 STA estimates for 1000 times and obtained the corresponding dynamic gain curves. The 95 percent confidence intervals from these bootstraps are shown for each dynamic gain curve, typically they are smaller than the line width.

We assessed the transfer impedance of stimuli transmitted from soma to the AP initiation site in purely passive neuron models, models, sodium peak conductance $\bar{g}_{Na}$ was set to 0. We injected the soma with sinusoidal stimuli of 1Hz to 1000Hz, and recorded voltage fluctuations along the axon. The transfer impedance function was calculated as the amplitude ratio of the Fourier transform of output voltage fluctuations to the Fourier transform of the input stimuli at corresponding frequencies.

## Supporting information

**S1 Fig. Axonal and somatic AP waveforms with various AP initiation sites.** AP waveforms are generated with constant inputs reproducing 5 Hz firing rate. Axonal and somatic voltages are reset to the resting potential when axonal voltages reach 0 mV.
(PDF)

**S2 Fig. Impact of AP initiation site on AP initiation dynamics and dynamic gain functions with various $k_a$. A, C, E** For each $k_a$, axonal voltage derivatives of three model variants aligned at (0 mV, 0 mV/ms). **B, D, F** Dynamic gain functions calculated with the firing rate fixed at 5 Hz, CV of ISI fixed at 0.85. $\tau$ is set to 5 and 50 ms for each model variant.
(PDF)

**S3 Fig. Brunel effect is larger when AP initiation dynamics is more voltage sensitive.** $x_{Na}$ is fixed at 40 $\mu$m. $k_a$ is set to 6 mV, 1 mV and 0.1 mV in A, B and C respectively. Increasing $\tau$ from 2 ms to 50 ms, the enhancement of dynamic gain in the high-frequency region is larger when $k_a$ is smaller. The same working point was used for all simulations (CV$_{ISI}$ = 0.85, $v$ = 5 Hz).
(PDF)

**S4 Fig. Myelination of the axon does not enhance the encoding bandwidth.** To simulate myelination of the axonal section, starting 60 $\mu$m from the soma, the specific membrane conductance was decreased 50 fold to $6.6^*10^{-7}$S/cm$^2$. The specific membrane capacitance was decreased 37.5 fold to 0.02$\mu$F/cm$^2$. $x_{Na}$ is 40$\mu$m. $k_a$ is 6mV. The firing rate is fixed at 5 Hz, and CV of ISI is 0.85. With a myelinated axon, the threshold is lower and therefore, AP initiation dynamics is slower (A and C). Bandwidth of the dynamic gain functions is hardly changed compared to those of the Brette's original model (B and D).
(PDF)

## Acknowledgments

We thank Barbara Feulner and Rainer Engelken for fruitful discussions.

## Author Contributions

**Conceptualization:** David Hofmann, Andreas Neef, Fred Wolf.

**Data curation:** Chenfei Zhang, David Hofmann.

**Funding acquisition:** Chenfei Zhang, Andreas Neef, Fred Wolf.

**Investigation:** Andreas Neef, Fred Wolf.

**Methodology:** Chenfei Zhang, David Hofmann, Andreas Neef, Fred Wolf.

**Resources:** Fred Wolf.

**Software:** Chenfei Zhang, David Hofmann, Andreas Neef, Fred Wolf.

**Supervision:** Andreas Neef, Fred Wolf.

**Validation:** Chenfei Zhang, Andreas Neef, Fred Wolf.

**Visualization:** Chenfei Zhang, David Hofmann, Andreas Neef.

**Writing – original draft:** Chenfei Zhang, David Hofmann, Fred Wolf.

**Writing – review & editing:** Chenfei Zhang, David Hofmann, Andreas Neef, Fred Wolf.

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
