## [Decision Letter · Decision Letter 0]

25 Oct 2021

Dear Dr. Neef,

Thank you very much for submitting your manuscript "Ultrafast population coding and axo-somatic compartmentalization" for consideration at PLOS Computational Biology.

As with all papers reviewed by the journal, your manuscript was reviewed by members of the editorial board and by several independent reviewers. In light of the reviews (below this email), we would like to invite the resubmission of a significantly-revised version that takes into account the reviewers' comments.

We cannot make any decision about publication until we have seen the revised manuscript and your response to the reviewers' comments. Your revised manuscript is also likely to be sent to reviewers for further evaluation.

Sincerely,

Michele Migliore

Associate Editor

PLOS Computational Biology

Lyle Graham

Deputy Editor

PLOS Computational Biology

Reviewer's Responses to Questions

**Comments to the Authors:**

Reviewer #1: This manuscript appears to be a sound study of the dynamic gain function of a neuron and the role of the axon initial segment. Brette and collaborators showed that the presence of axon initial segment results in a sharp spike initiation at the soma. The present manuscript shows that this does not imply high frequency spike rate transmission and that the latter depends of the voltage sensitivity of sodium channels. I think this is a welcome clarification.

My only reservation about the manuscript is that although it uses a reduced model, results are only obtained by numerical simulations. It would have been nice to support some of the conclusions by analytical results. I wonder whether this would be possible perhaps in an even simpler model, like a two-compartment soma-AIS model.

Reviewer #2: In this manuscript, Authors revisit and further analyse an existing model proposed by Brette and collaborators. They focus on geometry and biophysical properties of simplified, multicompartmental model, of a soma, an axon initial segment, and of an axon.

Authors study the response properties of the system in the context of the "dynamic gain", while carrying out exstensive numerical simulations.

While the manuscript has certainly value, I have a few concerns on methodological aspects that might have made the story much stronger.

The model is "simple" and not realistic. With abundance of multicompartmental, anatomically precise, biophysical models of cortical, hippocampal, and cerebellar neurons, I would have expected Authors to extend their reasoning and test their predictions and analysis also on more detailed models. The usefulness of simplified description is of course recognized but as numerical simulations and model data sharing are increasingly easy to perform nowadays, it seems that a more robust story could have been constructed.

Another point on the realism concerns the biophysical description of the action potential. The model considered by the Authors cannot produce "naturally" a train of action potentials. In fact, the lack of sodium channels inactivation and of repolarising currents, as explained in the Methods, is replaced by a hard "reset" of the membrane potential. While I understand that reducing the number of free parameters simplifies their systematic explorations, especially in the context of analytical exploration of a set of non-linear equations (e.g. at the steady-state), I fail to gain a convincing proof of the equivalence of the model response properties with and without the additional biophysical models.

Once more, as Authors have performed numerical explorations, they might have provided an equivalence of "full" and "reduced" models.

Another simplification is the use of current-based synaptic inputs versus conductance inputs. I believe it might be important to first prove their equivalence, in the specific context considered by the Authors. I fail to understand the advantages and opportunity in terms of simplicity of numerical simulations, while I believe that an activity-dependent "effective membrane time constant" may add additional realism to the Authors' point.

A final concern, related to the previous point, is the choice of the parameters of the OU process. If I am not mistaken, the "mu" and "sigma" should not be considered independent, as they have been. In the "conductance-driven" input recreation, I think "mu" and "sigma" will be both linked together. I understand the need of fixing the "operating point", in terms of firing frequency and coefficient of variation of the interspike interval distribution, but wonder whether the same operating point (e.g. in vivo high CV) would still be physiologically compatible with synaptic (conductance) stimulation.

Reviewer #3: Higher frequency sensitivity of neurons, exceeding the sensitivity expected from the passive membrane time constant, has previously been attributed to morphological adaptation, high channel densities, ion channel cooperativity and network properties. The authors of the manuscript argue that ion channel properties, in particular their voltage dependence, are more decisive than channel densities for high frequency sensitivity, priming the channel activation properties as a target for dynamical regulation of ultrafast population coding.

The topic is timely and of significant interest to the neuroscience community. However, the present paper strongly relies on a detailed understanding of Brette (2013). More care could be taken to ensure the manuscript is self-explanatory. Moreover, it would be important to clearly delineate the novelty in this manuscript compared to Brette (2013) or Eyal (2014) articles. In addition, there are crucial aspects regarding both techniques and presentation to be addressed in a revision of the manuscript.

More specific comments (in no particular order):

A more systematic exploration of how key features of the dynamic gain curve (cutoff and high frequency slope) vary with system parameters would be important.

Would a spatial EIF model (Aspart, Ladenbauer and Obermayer, PLoS CB, 2016) be able to reproduce the present results? Similar to a comparison in the original Fourcaud-Trocme et al (2003) paper.

The Brette paper discusses distributed sodium channels. How would that impact the model presented here?

What are the mean firing rates in the simulation? Make sure all parameter values are clear for each figure.

The model should be made available in an online repository. (l. 509) lists an old NEURON version. Please make sure that the simulations also run on the recent version (8.0).

(l. 51) Reference 23 is used twice in the sentence.

(l. 127) It would be worth mentioning the type of mathematical bifurcation that is referred to (a cusp, a saddle-node?). I is important to be precise as there are many different types of bifurcations all with different properties of which "sudden changes" is only one.

(Fig. 1B) It would be helpful to plot the distance $x_{Na}$ as labels into the subplots. The stable an unstable fixpoints should be marked as solid and empty dots.

(Fig. 1C) Label the "discontinuity" with a different line style or jump. There seems to be discrepancy between the main text referring to Figure and the caption. Is this figure for constant current or increasing current (voltage or current clamp, spiking with 5Hz or not)?

(l. 138) Although for a cusp bifurcation there is the possibility for hysteresis, it does not seem to be visible in Fig. 1C. Why?

(165) It would be worth restating the firing rate and CV values used also in this section.

(l. 167) With a cutoff of 200 Hz (tau = 5ms) it seems difficult to estimate the higher frequency gain. In the frequency range of interest > 10^2 Hz the input process has little power. This should reflect in the estimation errorbars of the spectral densities. Due to the large number of trials these are not visible. How where the errorbars calculated? Bootstrapping or error propagation? Why not use a faster time scale for the stimulus?

(l. 173) Could you draw the $f^{-1}$ line into the plot Fig. 2B. The model with $x_{Na}=0$ does not show a hight-frequency scaling exponent of -1. This seems to contradict the simulation of the isopotential WB model in Fourcaud-Trocme et al (2003). What is the exponent How can this be resolved?

(Fig. 2B) The number of spikes used to estimate the dynamic gain function (or the STA on which they are based?).

(l. 193) How exactly is the AP shape altered? Maybe a Figure in the appendix is appropriate?

(l. 195) It is unclear at this point what electrotonic filtering is. Is it the transfer impedance? Please elaborate. It is also not clear from the caption of Fig. 3.

(l. 210-214) Does this mean that the high-frequency dynamic gain varies non-monotonically with the position of the AP initiation site? If so, this is an interesting finding that could be highlighted.

(l. 216) Would the term axial resistivity be more appropriate?

(Fig. 5) The 3d surface plots with additional colour coding are difficult to read. Could you use simple heatmaps instead?

(l. 266) Could you explore more that two tau values? What happens for smaller values than 5ms? Does the weal resonance remain and at which frequency?

(l. 290) With only two values of ka it is difficult to see the general trend. Please increase the parameter set.

(l. 300-301) Is the 1/f scaling of the dynamic gain affected by ka?

(l. 305-307) The fact that an increase in xNa decreases the dynamic gain seem to be the opposite of what was seen in Fig. 2B, yet it is observed in Fig. 3C. What is the relation between Fig. 3C and Fig. 8?

(Fig. 8) The 3d surface plots with additional colour coding are difficult to read. Could you use simple heatmaps instead?

(l. 333-334) Not clear to which voltage clamp scenario in Fig. 1 is referred to.

(l. 337-338) Why would it behave like an LIF? Would the spike onset dynamics be irrelevant?

(l. 372-375) Could more extreme $x_{Na}$ cases be simulated to corroborate the claims here?

(l. 529) The prefactor to the second spatial derivatives in the cable equation (1) and (2) seems wrong. Should it be $\\frac{d}{4R_a}$?

(l. 534) Should the continuigy of the current boundary condition be $\\frac{d^2V}{dx^2}$?

(l. 552-554) It is not completely clear what the reset conditions are.

(l. 581) Eq. (4) does not have the firing rate removed from the Dirac delta train.

Reviewer #4: See attachment.

**Have the authors made all data and (if applicable) computational code underlying the findings in their manuscript fully available?**

Reviewer #1: **No: **The authors uses the publicly available neuron code and and provide the parameter they use.

Reviewer #2: Yes

Reviewer #3: **No: **See comments to authors.

Reviewer #4: None

PLOS authors have the option to publish the peer review history of their article (what does this mean?). If published, this will include your full peer review and any attached files.

Reviewer #1: No

Reviewer #2: No

Reviewer #3: No

Reviewer #4: **Yes: **Romain Brette
---

## [Decision Letter · Decision Letter 1]

16 Dec 2021

Dear Dr. Neef,

We are pleased to inform you that your manuscript 'Ultrafast population coding and axo-somatic compartmentalization' has been provisionally accepted for publication in PLOS Computational Biology.

Best regards,

Michele Migliore

Associate Editor

PLOS Computational Biology

Lyle Graham

Deputy Editor

PLOS Computational Biology

Reviewer's Responses to Questions

**Comments to the Authors:**

Reviewer #1: The authors have taken into account my comments and those of the other referees. Il think that the manuscript is now suitable for publication in pcbi.

Reviewer #3: The authors have addressed my concerns and I accept the counter arguments to some remarks.

**Have the authors made all data and (if applicable) computational code underlying the findings in their manuscript fully available?**

Reviewer #1: Yes

Reviewer #3: Yes

PLOS authors have the option to publish the peer review history of their article (what does this mean?). If published, this will include your full peer review and any attached files.

Reviewer #1: No

Reviewer #3: No

---

## [Editor Report · Acceptance letter]

6 Jan 2022

PCOMPBIOL-D-21-01639R1 

Ultrafast population coding and axo-somatic compartmentalization

Dear Dr Neef,

I am pleased to inform you that your manuscript has been formally accepted for publication in PLOS Computational Biology. Your manuscript is now with our production department and you will be notified of the publication date in due course.

With kind regards,

Olena Szabo
